# Blood pressure variability and early neurological deterioration according to the chronic kidney disease risk categories in minor ischemic stroke patients

**Jae-Chan Ryu**[1], **Jae-Han Bae**[1], **Sang Hee Ha**[1], **Jun Young Chang**[1], **Dong-Wha Kang**[1], **Sun U. Kwon**[1], **Jong S. Kim**[1], **Chung Hee Baek**[2], **Bum Joon Kim**[1]*

1 Department of Neurology, Asan Medical Center, University of Ulsan College of Medicine, Seoul, Korea,
2 Division of Nephrology, Department of Internal Medicine, Asan Medical Center, University of Ulsan College of Medicine, Seoul, Korea

* medicj80@hanmail.net

## Abstract

### Objective

Chronic kidney disease (CKD) increases blood pressure variability (BPV) and affects stroke outcomes. However, the effect of BPV on early neurological deterioration (END) may be different according to the renal function.

### Methods

We enrolled ischemic stroke patients with a National Institutes of Health Stroke Scale of ≤5. END was defined as worsening of ≥1 point in motor power or ≥2 points in total score. BPV was calculated with BP measured during the first 5 days and presented as standard deviation (SD) and coefficient of variation (CoV). Renal function was classified using the Kidney Disease Improving Global Outcomes (KDIGO) classification of CKD. Variables were compared between those with (KDIGO classification: moderate- to very-high-risk) and without renal impairment (KDIGO classification: low-risk) and factors associated with END were investigated.

### Results

Among the 290 patients (136 [46.9%] renal impairment), END was observed in 59 (20.3%) patients. BPV parameters and the risk of END increased as renal function was impaired. Renal function and systolic BP (SBP) mean, SD, CoV, and diastolic BP (DBP) mean, SD were independently associated with END. We found no association between BPV parameters and END in normal renal function patients; however, among impaired renal function patients, SBP SD (odds ratio [OR]: 1.20, 95% confidence interval [CI]: 1.09–1.32, *P*<0.001) and CoV (1.30 [1.12–1.50], *P*<0.001) were associated with END.

**Data Availability Statement:** All relevant data are within the paper and its Supporting information files.

**Funding:** This research was supported by grants from the Brain Convergence Research Program of the National Research Foundation funded by the Korean government (MSIT No. 2020M3E5D2A01084576) and the National Research Foundation of Korea (NRF)grant funded by the Korea government (MSIT No. 2020R1A2C2100077). The funder had no role in study design, data collection ana analysis, decision to publish, or preparation of the manuscript.

**Competing interests:** The authors have declared that no competing interests exist.

## Conclusions

The association between END and BPV parameters differs according to renal function in minor ischemic stroke; BPV was associated with END in patients with renal impairment, but less in those with normal renal function.

## Introduction

Chronic kidney disease (CKD) is a well-known independent risk factor for cerebrovascular diseases, including ischemic stroke [1–3]. Moreover, the functional outcomes of ischemic stroke can be affected by the presence and severity of CKD [4]. The severity of CKD is defined by the estimated glomerular filtration rate (eGFR), which represents the residual renal function, and the severity of proteinuria, which is the result of increased permeability of the damaged capillary wall and impaired resorption. Both decreased eGFR and presence of proteinuria are independently associated with the outcomes of stroke [5, 6]. In addition to the vascular risk factors including aging, hypertension, and diabetes, the damage to the brain and kidney share similar pathomechanisms that affect microvasculature due to anatomical similarities [7].

Early neurological deterioration (END) is defined as neurological worsening during the acute stage, which influences stroke outcomes, especially in initially minor strokes [8, 9]. The presence of CKD has been thought to increase the risk of END. Hypothetically, endothelial dysfunction, chronic inflammation, and oxidative stress have been regarded as mechanisms of CKD that affect neurological deterioration [10, 11]. On the other hand, blood pressure variability (BPV) is associated with arterial compliance, and affects cerebral microcirculation and blood-brain barrier [12, 13]. Previous studies showed that increased BPV in the acute stage of stroke has also been associated with an increased risk for END and poor outcomes of stroke [14, 15]. Patients with reduced renal function show increased BPV [16, 17].

Based on previous studies, we hypothesized that patients with ischemic stroke with impaired renal function may show a higher BPV, and that BPV may be associated with END and the effect of BPV on END may differ according to renal function. For verification, we investigated CKD as a risk factor for END and the effects of BPV in minor ischemic stroke patients with and without renal impairment based on the Kidney Disease Improving Global Outcomes (KDIGO) classification of CKD.

## Methods

### Participants

We retrospectively reviewed the data from the patients with acute minor ischemic stroke who were admitted to the Asan Medical Center between October 2019 and May 2020. Patients were included in our study if they fulfilled the following criteria: (1) age ≥18 years; (2) time from symptom onset to hospital admission of ≤7 days; (3) acute ischemic stroke confirmed with a diffusion-weighted image, and (4) minor stroke defined with a National Institutes of Health Stroke Scale (NIHSS) score of ≤5. We excluded patients with incomplete medical histories and who had end-stage renal disease (ESRD) including those who were on hemodialysis because hemodialysis can cause changes in blood pressure (BP).

We determined the stroke subtypes using the Trial of Org 10172 in Acute Stroke Treatment (TOAST) subtype classification system: large-artery atherosclerosis, cardioembolic stroke, small-vessel occlusion, undetermined, and other determined. Informed consent from the

patients was not obtained because the study was retrospective. The local ethics committee of Asan Medical Center approved this study (IRB No. 2021–1269).

## Renal function, BPV, and END

KDIGO guidelines classify CKD into four groups according to eGFR and albuminuria [18]. eGFR was determined using the CKD−EPI equation [19]. Creatinine level and eGFR were measured on the first admission day in the emergency department. Moreover, albuminuria was estimated as the urine albumin/creatinine ratio obtained from spot urine analysis on the first admission day; normoalbuminuria was indicated by <30 mg/g of creatinine, microalbuminuria by 30−300 mg/g of creatinine, and macroalbuminuria by >300 mg/g of creatinine. In KDIGO classification of CKD, decreased eGFR and increased albuminuria are associated with increased risk of adverse outcomes including CKD progression, ESRD, cardiovascular disease, and mortality. The risk of adverse outcomes is classified in four groups: low, moderate, high, and very high risk group. Low-risk group is defined as eGFR ≥60 ml/min/1.73m$^2$ with normoalbuminuria, moderate-risk group as 1) eGFR ≥60 ml/min/1.73m$^2$ with microalbuminuria or 2) 45−59 ml/min/1.73m$^2$ of eGFR with normoalbuminuria, High-risk group is defined as 1) 30−44 ml/min/1.73m$^2$ with normoalbuminuria, 2) 45−59 ml/min/1.73m$^2$ with microalbuminruia, or 3) eGFR ≥60 ml/min/1.73m$^2$ with macroalbuminuria. Finally, very-high-risk group is defined as 1) eGFR <30 ml/min/1.73m2 with normoalbuminuria, 2) eGFR <45 ml/min/1.73m2 with microalbuminuria, or 3) eGFR <60 ml/min/1.73m2 with macroalbuminuria. Moreover, the low-risk group is recognized as having normal renal function in KDIGO classification. Therefore, we divided study population into the two groups: normal renal function group (low-risk group) and impaired renal function group (moderate, high, and very-high-risk group).

BP was measured using a validated, calibrated, automatic, and noninvasive BP-monitoring device (IntelliVue MP50; Philips MedizinSysteme, Böblingen, Germany) in the acute stroke unit and general ward according to our local stroke center's protocol as follows; all BP measurements were performed in a resting, comfortable state with quiet environment; during the patient's stay in the acute stroke unit, BP was regularly measured every 6 hours; in the general ward, every 8 hours, regardless of day and night; permissive BP is allowed and patients with anti-hypertensive medications stopped taking anti-hypertensive medications in the acute stage of ischemic stroke. In our analysis, we used the BP recorded during the first 5 days of hospitalization. We excluded BP data after END because these could be confounded by additional factors, such as induced hypertension treatment. We calculated the systolic blood pressure (SBP) variability and diastolic blood pressure (DBP) variability, presenting them as standard deviations (SDs) and coefficient of variation (CoV; equal to [SD × 100]/mean) [14]. Additionally, average real variability (ARV) was also calculated. We calculated BPV of each subject, and then calculated the mean of the variability according to the group.

Severity of the stroke was determined using the NIHSS score, which was evaluated by trained nurse (every 4 hours in acute stroke unit and every 8 hours in general ward) and confirmed by a neurologist. END was defined as an increase of at least 1 point in motor power or a total NIHSS score deterioration of ≥2 points within 3 days after admission [8]. The class of BP lowering medication and antithrombotic agent before admission were also investigated.

## Statistical analysis

First, we compared the baseline characteristics and the presence of END in the four risk groups divided according to the KDIGO classification of CKD. The significance of the intergroup differences was assessed using chi-square test, Kruskal-Wallis test, and one way ANOVA. Then,

we compared the characteristics of the patients with and without END. In this comparison, the significance of the intergroup differences was assessed using chi-square test, Mann–Whitney U test, and Student's *t* test, as appropriate. Using the multivariable logistic regression model, we analyzed the independent factors associated with END, including those from the univariate model. Thereafter, the association between BPV and END in patients with the normal renal function group (low-risk group) versus the impaired renal function group (moderate- to very high risk groups) were investigated. Finally, P for interactions between renal function and BPV parameters for the occurrence of END were analyzed. All analyses were performed using R Statistical Software (version 4.0.5; R Foundation for Statistical Computing, Vienna, Austria).

## Results

During the study period, 635 patients were admitted to our center for ischemic stroke within 7 days from stroke onset. Of these, 321 patients (50.6%) were classified as having a minor ischemic stroke. We excluded 15 patients with insufficient medical histories, and 16 patients who were receiving hemodialysis. Thus, we included 290 patients in the final analysis (S1 Fig).

The mean age of the enrolled patients was 67.0 ± 12.7 years-old, and 183 (63.1%) were men. Among these, 136 (46.9%) showed impaired renal function (moderate, high, and very high risk) and 59 (20.3%) had experienced END. In patients without END, the BP was measured 12.0 ± 0.2 times on average, and in those with END, the BP was measured 9.5 ± 1.6 times on average ($P < 0.001$).

### Patient characteristics according to KDIGO classification

According to the KDIGO classification of CKD, 154 (53.1%) patients were at low risk and 97 (33.4%), 22 (7.6%), and 17 (5.9%) patients were at moderate, high, and very high risk (impaired renal function), respectively. The clinical characteristics according to KDIGO classification are summarized in Table 1. The mean age significantly increased with the increased risk assessed by KDIGO classification ($P < 0.001$). The prevalence of hypertension, diabetes, and history of coronary artery disease also increased ($P = 0.002$, $P < 0.001$, and $P = 0.008$, respectively). There were also significant differences in neurological severity at admission and on discharge. ($P = 0.025$ and $P < 0.001$, respectively). However, the stroke subtypes did not differ according to KDIGO classification. There were no differences in the use of anti-hypertensive medication and antithrombotic agent before admission according to the risk of KDIGO classification.

Moreover, SBP mean, SD, and DBP SD, CoV increased as the risk estimated by KDICO classification increased ($P = 0.002$, $P = 0.040$, $P = 0.004$, and $P = 0.002$, respectively). The prevalence of END increased in the very-high-risk group (47.1%) compared with the low-risk group (11%; $P < 0.001$; Table 1). The distribution of patients according to the KDIGO classification are described in Fig 1.

### Prognostic factors of END

Patients with END were older (66.0 ± 12.8 vs. 71.0 ± 11.5 years; $P = 0.007$). No differences existed for risk factors or stroke subtypes between those with and without END. The distribution of adverse outcome risk according to KDIGO classification was higher ($P < 0.001$), and the prevalence of albuminuria was more frequent in patients with END compared with in those without ($P < 0.001$). In those with END, the SBP mean, SD, CoV, and DBP mean, SD were higher ($P = 0.002$, $P < 0.001$, $P = 0.003$, $P = 0.023$, and $P = 0.006$, respectively; Table 2).

Univariate analysis showed that old age (odds ratio [OR]: 1.04; 95% confidence interval [CI]: 1.01–1.06, $P = 0.008$), higher admission NIHSS (1.19 [1.01–1.41], $P = 0.043$), SBP mean

**Table 1. Comparison of patient clinical characteristics and BPV according to the risk of KDIGO classification.**

| Characteristic | The risk of KDIGO classification | | | | P value |
|---|---|---|---|---|---|
| | Low (N = 154) | Moderate (N = 97) | High (N = 22) | Very high (N = 17) | |
| Age, years | 63.1 ± 12.7 | 69.8 ± 11.7 | 75.4 ± 7.5 | 75.2 ± 9.9 | <0.001 |
| Male | 98 (63.6) | 62 (63.9) | 11 (50.0) | 12 (70.6) | 0.556 |
| Vascular risk factor | | | | | |
| Hypertension | 98 (63.6) | 73 (75.3) | 21 (95.5) | 15 (88.2) | 0.002 |
| Diabetes mellitus | 31 (20.1) | 37 (38.1) | 12 (54.5) | 8 (47.1) | <0.001 |
| Hyperlipidemia | 70 (45.5) | 53 (54.6) | 14 (63.6) | 9 (52.9) | 0.285 |
| Atrial fibrillation | 16 (10.4) | 20 (20.6) | 5 (22.7) | 2 (11.8) | 0.082 |
| CAD | 16 (10.4) | 18 (18.6) | 3 (13.6) | 7 (41.2) | 0.008 |
| Current smoking | 61 (39.6) | 33 (34.0) | 8 (36.4) | 10 (58.8) | 0.272 |
| Stroke history | 40 (26.0) | 31 (32.0) | 9 (40.9) | 6 (35.3) | 0.417 |
| Laboratory findings | | | | | |
| HbA1c, % | 5.9 ± 0.9 | 6.5 ± 1.4 | 6.4 ± 1.1 | 6.8 ± 2.0 | <0.001 |
| LDL, mg/dL | 112 ± 40 | 112 ± 45 | 105 ± 43 | 99 ± 43 | 0.413 |
| r-tPA | 5 (3.2) | 6 (6.2) | 1 (4.5) | 0 (0.0) | 0.552 |
| Admission NIHSS | 2 [1−3] | 3 [1−4] | 3 [1−5] | 1 [0−3] | 0.025 |
| Discharge NIHSS | 1 [0−3] | 3 [1−5] | 2 [1−4] | 3 [2−5] | <0.001 |
| END | 17 (11.0) | 30 (30.9) | 4 (18.2) | 8 (47.1) | <0.001 |
| TOAST classification | | | | | 0.778 |
| LAD | 38 (24.7) | 27 (27.8) | 4 (18.2) | 4 (23.5) | |
| SVO | 61 (39.6) | 28 (28.9) | 6 (27.3) | 5 (29.4) | |
| CE | 24 (15.6) | 19 (19.6) | 5 (22.7) | 4 (23.5) | |
| UD | 22 (14.3) | 18 (18.6) | 6 (27.3) | 2 (11.8) | |
| OD | 9 (5.8) | 5 (5.2) | 1 (4.5) | 2 (11.8) | |
| BPV | | | | | |
| SBP mean, mmHg | 136.5 ± 17.7 | 139.8 ± 19.1 | 150.6 ± 18.4 | 144.0 ± 23.4 | 0.002 |
| SBP SD | 11.1 ± 4.8 | 11.7 ± 4.9 | 13.2 ± 4.2 | 12.7 ± 4.7 | 0.040 |
| SBP CoV | 8.2 ± 3.6 | 8.4 ± 3.4 | 8.7 ± 2.4 | 8.8 ± 2.7 | 0.382 |
| DBP mean, mmHg | 78.8 ± 10.6 | 79.3 ± 11.3 | 82.8 ± 8.8 | 74.8 ± 19.9 | 0.883 |
| DBP SD | 7.2 ± 2.4 | 8.2 ± 3.1 | 7.9 ± 2.8 | 8.8 ± 4.0 | 0.004 |
| DBP CoV | 9.2 ± 3.2 | 10.6 ± 4.2 | 9.5 ± 3.3 | 12.1 ± 5.2 | 0.002 |
| BP lowering agent at admission | 72 (46.8) | 55 (56.7) | 18 (81.8) | 14 (82.4) | |
| RAS inhibitors | 54 (75.0) | 37 (67.3) | 10 (55.6) | 7 (50.0) | 0.172 |
| β-blockers | 9 (12.5) | 12 (21.8) | 3 (16.7) | 5 (35.7) | 0.175 |
| CCBs | 43 (59.7) | 38 (69.1) | 12 (66.7) | 8 (57.1) | 0.681 |
| Diuretics | 17 (23.6) | 8 (14.5) | 2 (11.1) | 2 (14.3) | 0.448 |
| Antithrombotic agent at admission | 54 (35.1) | 47 (48.5) | 13 (59.1) | 12 (70.6) | 0.801 |
| Antiplatelet agent | 39 (72.2) | 39 (83.0) | 9 (69.2) | 10 (83.4) | |
| Anticoagulation | 12 (22.2) | 6 (12.8) | 3 (23.1) | 1 (8.3) | |
| Both | 3 (5.6) | 2 (4.2) | 1 (7.7) | 1 (8.3) | |

Values are expressed as number (% column), mean ± standard deviation or median (interquartile range).
KDIGO, Kidney Disease Improving Global Outcome; BPV, blood pressure variability; CKD, chronic kidney disease; CAD, coronary artery disease; HbA1c, hemoglobin A1c; LDL, low-density lipoprotein; r-tPA, recombinant tissue plasminogen activator; NIHSS, National Institutes of Health Stroke Scale; END, early neurological deterioration; TOAST, Trial of Org 10172 in Acute Stroke Treatment; LAD, large artery disease; SVO, small-vessel occlusion; CE, cardioembolism; UD, undetermined cause; OD, other determined cause; SBP, systolic blood pressure; DBP, diastolic blood pressure; SD, standard deviation; CoV, coefficient of variation; RAS, renin-angiotensin system; CCBs, calcium channel blockers.

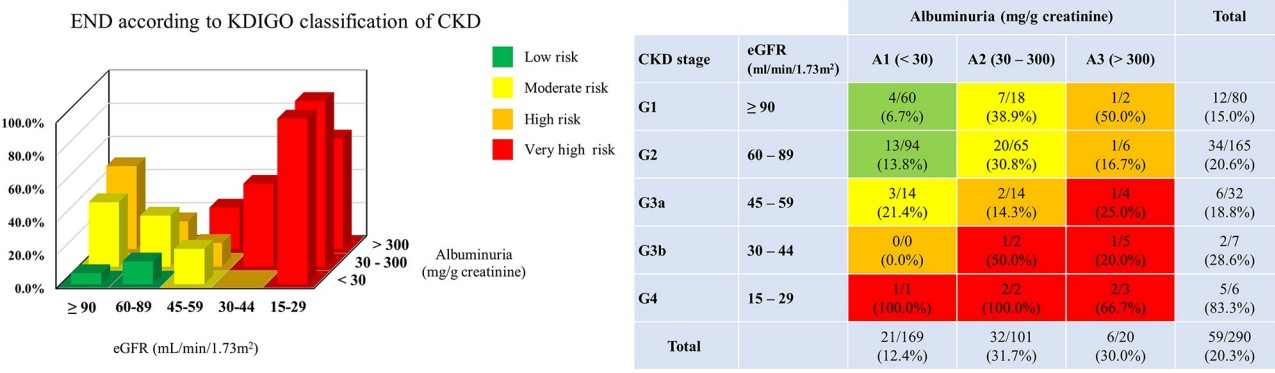

| CKD stage | eGFR (ml/min/1.73m²) | Albuminuria (mg/g creatinine) | | | Total |
|---|---|---|---|---|---|
| | | A1 (< 30) | A2 (30 − 300) | A3 (> 300) | |
| G1 | ≥ 90 | 4/60 (6.7%) | 7/18 (38.9%) | 1/2 (50.0%) | 12/80 (15.0%) |
| G2 | 60 − 89 | 13/94 (13.8%) | 20/65 (30.8%) | 1/6 (16.7%) | 34/165 (20.6%) |
| G3a | 45 − 59 | 3/14 (21.4%) | 2/14 (14.3%) | 1/4 (25.0%) | 6/32 (18.8%) |
| G3b | 30 − 44 | 0/0 (0.0%) | 1/2 (50.0%) | 1/5 (20.0%) | 2/7 (28.6%) |
| G4 | 15 − 29 | 1/1 (100.0%) | 2/2 (100.0%) | 2/3 (66.7%) | 5/6 (83.3%) |
| Total | | 21/169 (12.4%) | 32/101 (31.7%) | 6/20 (30.0%) | 59/290 (20.3%) |

**Fig 1. Distribution of the percentage of END patients according to KDIGO classification.** Low-risk group is green, moderate-risk group is yellow, high-risk group is dark orange, and very-high-risk group is red. CKD stage 5 (eGFR <15 mL/min/1.73m²) was not presented in this figure, since all patients with CKD stage 5 were on hemodialysis. END, early neurological deterioration; KDIGO, Kidney Disease Improving Global Outcome; CKD, chronic kidney disease.

(1.02 [1.01–1.04], $P = 0.002$), SBP SD (1.12 [1.06–1.19]. $P < 0.001$), SBP CoV (1.13 [1.04–1.22], $P = 0.005$), DBP mean (1.03 [1.00–1.06], $P = 0.019$), and KDIGO classification (reference—low risk; moderate risk—3.61 [1.86–7.00]; $P < 0.001$; very high risk—7.16 [2.44–21.04]; $P < 0.001$) were associated with END.

In multivariable logistic analysis for each BPV parameters, SBP mean (1.02 [1.00–1.04], $P = 0.013$), SBP SD (1.13 [1.06–1.20], $P < 0.001$), SBP CoV (1.14 [1.04–1.24], $P = 0.004$), and DBP mean (1.04 [1.01–1.07], $P = 0.004$) were associated with the presence of END. Moreover, the risk of KDIGO classification, especially moderate and very high risk group, was independently associated with the presence of END (S1 Table).

## BP and END according to renal function

For the patients with normal renal function (KDIGO classification; low risk), only SBP mean showed significant difference between those with and without END ($P = 0.038$). However, for the patients with impaired renal function (KDIGO classification: moderate to very high risk), SBP SD, SBP CoV, and DBP mean were higher in those with END than in those without ($P < 0.001$, $P = 0.001$, and $P = 0.026$, respectively; Fig 2).

According to the renal function, we separately adjusted the BP and BPV parameters for the potential factors ($P < 0.20$) in univariable analysis (Table 3), finding no association between BPV parameters and END among those patients with normal renal function. In contrast, among those with impaired renal function, SBP SD (1.20 [1.09–1.32], $P < 0.001$), SBP CoV (1.30 [1.12–1.50], $P < 0.001$), and DBP mean (1.04 [1.00–1.07], $P = 0.027$) were associated with END. Additionally, logistic regression analysis of SBP and DBP ARV were also performed (S2 Table). Although SBP ARV was associated with END in both groups, DBP ARV was only associated with END in impaired renal function group. Finally, in the analysis of $P$ for interaction between renal function and BPV parameters for the occurrence of END, SBP CoV (1.17 [0.98–1.41], $P = 0.085$) and DBP SD (1.12 [0.99–1.26], $P = 0.079$) approached the borderline of significance (S3 Table).

## Discussion

The study shows that the presence of END, BP, and BPV parameters can vary depending on KDIGO classification of CKD; as the risk based on KDIGO classification increased, the BPV parameters and END prevalence increased. Among those patients classified as very high risk

**Table 2. Comparison of patient clinical characteristics and BPV in the groups with and without END.**

| Characteristic | Non-END (N = 231) | END (N = 59) | P value |
|---|---|---|---|
| Age, years | 66.0 ± 12.8 | 71.0 ± 11.5 | 0.007 |
| Male sex | 143 (61.9) | 40 (67.8) | 0.493 |
| Vascular risk factor | | | |
| Hypertension | 161 (69.7) | 46 (78.0) | 0.274 |
| Diabetes mellitus | 67 (29.0) | 21 (35.6) | 0.410 |
| Hyperlipidemia | 120 (51.9) | 26 (44.1) | 0.350 |
| Atrial fibrillation | 34 (14.7) | 9 (15.3) | >0.999 |
| CAD | 32 (13.9) | 12 (20.3) | 0.300 |
| Current smoking | 90 (39.0) | 22 (37.3) | 0.932 |
| Stroke history | 66 (28.6) | 20 (33.9) | 0.522 |
| Laboratory findings | | | |
| HbA1c, % | 6.2 ± 1.2 | 6.4 ± 1.3 | 0.272 |
| eGFR, mL/min/1.73m$^2$ | 79.8 ± 18.2 | 73.0 ± 21.8 | 0.092 |
| Albuminuria, mg/g creatinine | 15.2 [6.3–52.0] | 41.6 [13.8–120.3] | <0.001 |
| KDIGO classification | | | <0.001 |
| Low risk | 137 (59.3) | 17 (28.8) | |
| Moderate risk | 67 (29.0) | 30 (50.8) | |
| High risk | 18 (7.8) | 4 (6.8) | |
| Very high risk | 9 (3.9) | 8 (13.6) | |
| r-tPA | 8 (3.5) | 4 (6.8) | 0.438 |
| Admission NIHSS | 2 [1–4] | 3 [1–4] | 0.038 |
| Discharge NIHSS | 1 [0–3] | 5 [4–7] | <0.001 |
| TOAST classification | | | 0.745 |
| LAD | 58 (25.1) | 15 (25.4) | |
| SVO | 78 (33.8) | 22 (37.3) | |
| CE | 44 (19.0) | 8 (13.6) | |
| UD | 39 (16.9) | 9 (15.3) | |
| OD | 12 (5.2) | 5 (8.5) | |
| SBP mean, mmHg | 137.3 ± 18.0 | 145.9 ± 21.0 | 0.002 |
| SBP SD | 11.0 ± 4.6 | 13.8 ± 4.8 | <0.001 |
| SBP CoV | 8.1 ± 3.4 | 9.5 ± 3.1 | 0.003 |
| DBP mean, mmHg | 78.2 ± 10.3 | 82.4 ± 13.0 | 0.023 |
| DBP SD | 7.4 ± 2.6 | 8.7 ± 3.4 | 0.006 |
| DBP CoV | 9.6 ± 3.8 | 10.6 ± 3.8 | 0.071 |

Values are expressed as number (% column), mean ± standard deviation, or median (interquartile range). In patients without END, the BP was measured 12 times on average, and in those with END, the BP was measured 10 times on average.

BPV, blood pressure variability; END, early neurological deterioration; CAD, coronary artery disease; HbA1c, hemoglobin A1c; LDL, low-density lipoprotein; eGFR, estimated glomerular filtration rate; r-tPA, recombinant tissue plasminogen activator; NIHSS, National Institutes of Health Stroke Scale; TOAST, Trial of Org 10172 in Acute Stroke Treatment; LAD, large artery disease; SVO, small-vessel occlusion; CE, cardioembolism; UD, undetermined cause; OD, other determined cause; SBP, systolic blood pressure; DBP, diastolic blood pressure; SD, standard deviation; CoV, coefficient of variation

in our study, a considerable proportion of patients showed END. Furthermore, KDIGO classification and BPV were independent factors affecting END. However, the effect of BPV was different according to renal function. Note that BPV parameters were not significantly associated with END in the normal renal function group, whereas in the impaired renal function group, SBP SD, SBP CoV, and DBP mean were independently associated with END.

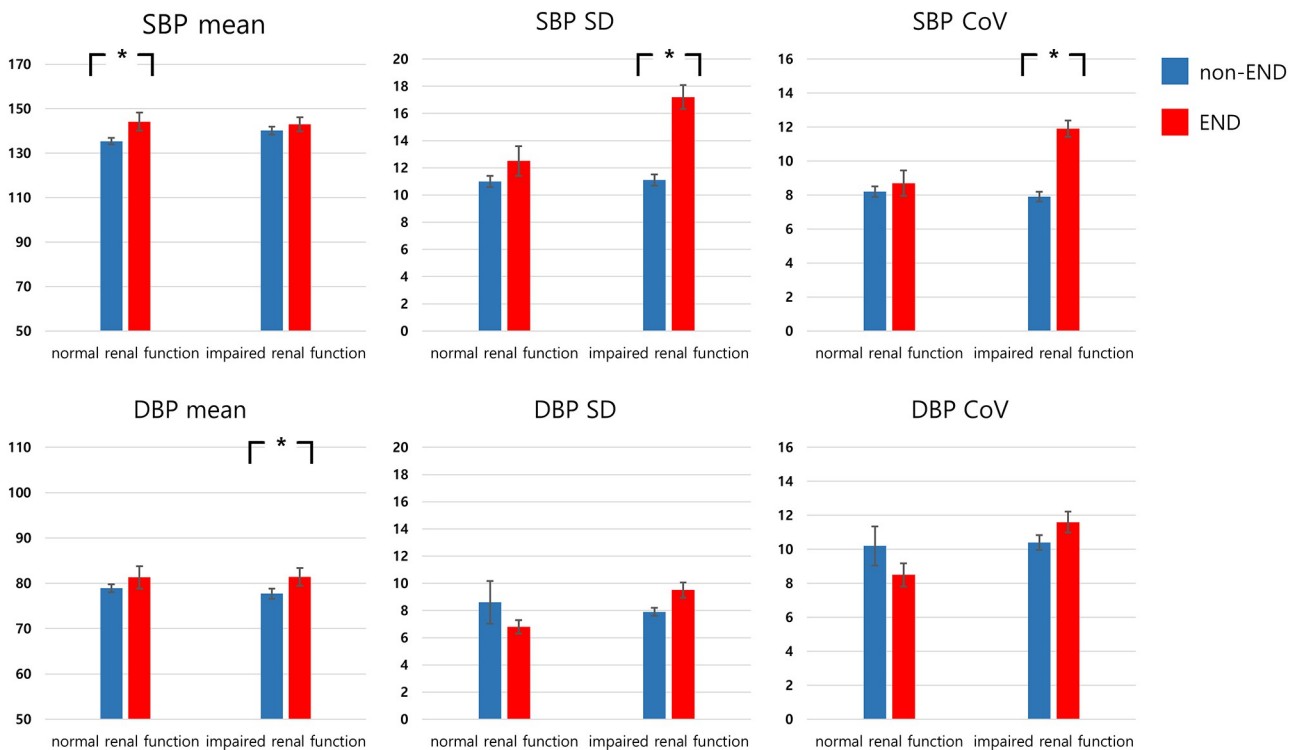

**Fig 2. Comparison of BPV parameters according to the presence of END and renal function.** BPV, blood pressure variability; END, early neurological deterioration; SBP, systolic blood pressure; DBP, diastolic blood pressure; SD, standard deviation; CoV, coefficient of variation.*Statistical significance: $P < 0.05$.

Previous studies have shown that various CKD parameters, such as eGFR, albuminuria, and cystatin C, were associated with the outcomes of stroke [5, 6, 20]. Reduced eGFR well represented overall kidney function, whereas albuminuria could show the extent of endothelial damage of kidney. The two parameters were complementary for predicting renal outcomes and both well predicted the progression to end-stage renal disease; the presence of albuminuria has been associated with END in subcortical infarction by showing infarct volume expansion,

**Table 3. Logistic regression analysis of BPV parameter as predictors for END according to renal function.**

| Variables | Normal renal function (N = 154) | P value | Impaired renal function (N = 136) | P value |
|---|---|---|---|---|
| | Adjusted OR (95% CI) | | Adjusted OR (95% CI) | |
| SBP mean, mmHg | 1.02 (0.99–1.06) | 0.106 | 1.02 (1.00–1.04) | 0.062 |
| SBP SD* | 1.06 (0.96–1.19) | 0.254 | 1.20 (1.09–1.32) | <0.001 |
| SBP CoV* | 1.08 (0.93–1.25) | 0.298 | 1.30 (1.12–1.50) | <0.001 |
| DBP mean, mmHg | 1.02 (0.98–1.07) | 0.338 | 1.04 (1.00–1.07) | 0.027 |
| DBP SD* | 0.99 (0.94–1.03) | 0.527 | 1.13 (0.99–1.28) | 0.064 |
| DBP CoV* | 0.98 (0.93–1.04) | 0.517 | 1.09 (0.99–1.20) | 0.091 |

SBP and DBP mean were adjusted for were adjusted for potential factors (P <0.20) for END. In normal renal function group, age and admission NIHSS were adjusted. On the other hand, sex, and hyperlipidemia were adjusted in impaired renal function group.

*SBP SD and CoV were additionally adjusted for SBP mean, and DBP SD and CoV were additionally adjusted for DBP mean.

BPV, blood pressure variability; END, early neurological deterioration; OR, odds ratio; CI, confidence interval; SBP, systolic blood pressure; DBP, diastolic blood pressure; SD, standard deviation; CoV, coefficient of variation

[11] and eGFR has been associated with functional outcome after acute ischemic stroke [21]. KDIGO classification of CKD encompasses eGFR and albuminuria, which is the two complementary and widely used predictors for CKD, and was highly associated with END.

CKD increases BPV by sympathetic overactivity, reduced arterial compliance, and fluctuation of the renin angiotensin aldosterone system [12]. Daily increased BPV after the index stroke, in the acute stage, may influence the occurrence of END [22]. Real-time hemodynamic alterations can influence the perfusion state, leading to infarction growth in the unstable phase of acute stroke. BPV in the acute phase of stroke was again associated with END in our current study. However, it was more associated with END among those with impaired renal function.

The damage to the microvasculature in the brain and kidneys correlate to each other, as both have shown anatomical similarities; the low resistance and sudden decrease in vessel diameter of the glomerulus and cerebral perforators can lead to a high correlation between impaired renal function and imaging biomarkers of damage to the cerebral microvasculature [11, 13]. Patients with ischemic stroke presenting with these biomarkers were also prone to END. Moreover, the uremic toxins can directly increase oxidative stress, endothelial dysfunction, and proinflammatory conditions enhancing neuronal death, leading to END among patients with impaired renal function [23]. Therefore, the microvascular fragility of the brain may at least partially explain the high risk of END among those with impaired renal function. According to our findings, we also can add another factor—BPV—to explain the high rate of END among patients with impaired renal function. The chronically increased BPV in patients with CKD may have compromised cerebral autoregulation, leading to a further progressive microvascular damage and increasing the potential for END. Finally, BPV in the acute phase may have influenced the fragile microcirculation increasing the risk of END.

Our study has some limitations. First, because we performed the study at a single center with a small sample size, it is difficult to generalize the results. Especially, the number of high risk and very high risk group was too small. Therefore, the findings require further verification in a larger, prospective study. However, measurement of BP was performed using the center's protocol, which was standardized across patients. Second, we measured albuminuria and creatinine only once on the day of admission. These parameters can fluctuate according to the sampling time or stress of the stroke. Follow-up data for albuminuria and creatinine may have strengthened our results. Third, it is well known that obesity has been associated with low-grade false positive albuminuria. Additional test for the obese patients would have improved our study more clearly [24].

Despite these limitations, we have shown that the effects of BPV on END are associated with renal function in acute minor ischemic stroke. BPV and END increased as the renal function decreased according to KDIGO classification. BPV was associated with END in patients with impaired renal function, but less in those with normal renal function. Therefore, we must consider BPV more carefully in patients with impaired renal function.

## Supporting information

**S1 Table. Multivariable logistic regression analysis of predictors for END in minor ischemic stroke patients.**
(DOCX)

**S2 Table. Logistic regression analysis of SBP and DBP ARV as predictors for END according to renal function.**
(DOCX)

**S3 Table. Interaction between renal function and BPV parameters for the occurrence of END.**
(DOCX)

**S4 Table. Database containing patient information.**
(PDF)

**S1 Fig. Study flow chart.** NIHSS, National Institutes of Health Stroke Scale.
(TIF)

## Author Contributions

**Conceptualization:** Jae-Chan Ryu, Bum Joon Kim.

**Data curation:** Jae-Chan Ryu, Jae-Han Bae, Sang Hee Ha, Jun Young Chang, Dong-Wha Kang, Sun U. Kwon, Jong S. Kim, Chung Hee Baek, Bum Joon Kim.

**Formal analysis:** Jae-Chan Ryu, Bum Joon Kim.

**Investigation:** Jae-Chan Ryu.

**Methodology:** Jae-Chan Ryu, Jae-Han Bae, Sang Hee Ha, Jun Young Chang, Dong-Wha Kang, Sun U. Kwon, Jong S. Kim, Chung Hee Baek, Bum Joon Kim.

**Supervision:** Bum Joon Kim.

**Writing – original draft:** Jae-Chan Ryu, Bum Joon Kim.

**Writing – review & editing:** Jae-Chan Ryu, Jae-Han Bae, Sang Hee Ha, Jun Young Chang, Dong-Wha Kang, Sun U. Kwon, Jong S. Kim, Chung Hee Baek, Bum Joon Kim.

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
