## [Decision Letter · Decision Letter 0]

2 Aug 2022

PONE-D-22-18870Blood Pressure Variability and Early Neurological Deterioration according to Renal Function in Minor Ischemic Stroke PatientsPLOS ONE

Dear Dr. Kim, 

Thank you for submitting your manuscript to PLOS ONE. After careful consideration, we feel that it has merit but does not fully meet PLOS ONE’s publication criteria as it currently stands. Therefore, we invite you to submit a revised version of the manuscript that addresses the points raised during the review process. Please submit your revised manuscript by September 15, 2022. If you will need more time than this to complete your revisions, please reply to this message or contact the journal office at plosone@plos.org. Please include the following items when submitting your revised manuscript:A rebuttal letter that responds to each point raised by the academic editor and reviewer(s). You should upload this letter as a separate file labeled 'Response to Reviewers'.A marked-up copy of your manuscript that highlights changes made to the original version. You should upload this as a separate file labeled 'Revised Manuscript with Track Changes'.An unmarked version of your revised paper without tracked changes. You should upload this as a separate file labeled 'Manuscript'.

We look forward to receiving your revised manuscript.

Kind regards,

Donovan Anthony McGrowder, PhD., MA., MSc

Academic Editor

PLOS ONE

Journal Requirements:

2. In the ethics statement in the Methods and online submission information, please ensure that you have specified (1) whether consent was informed and (2) what type you obtained (for instance, written or verbal, and if verbal, how it was documented and witnessed). If your study included minors, state whether you obtained consent from parents or guardians. If the need for consent was waived by the ethics committee, please include this information.

Additional Editor Comments:

Dear Dr. Kim,

Your manuscript “ Blood Pressure Variability and Early Neurological Deterioration according to Renal Function in Minor Ischemic Stroke Patients” has been assessed by our reviewers. They have raised a number of points which we believe would improve the manuscript and may allow a revised version to be published in PLOS ONE. Their reports, together with any other comments, are below.

If you are able to fully address these points, we would encourage you to submit a revised manuscript to PLOS ONE.

Kind regards,

Donovan Anthony McGrowder

Academic Editor

PLOS ONE

Reviewers' comments:

Reviewer's Responses to Questions

**Comments to the Author**

1. Is the manuscript technically sound, and do the data support the conclusions?

Reviewer #1: Partly

Reviewer #2: Yes

Reviewer #3: Yes

Reviewer #4: Partly

Reviewer #5: Yes

Reviewer #6: Partly

2. Has the statistical analysis been performed appropriately and rigorously? 

Reviewer #1: No

Reviewer #2: Yes

Reviewer #3: Yes

Reviewer #4: I Don't Know

Reviewer #5: Yes

Reviewer #6: No

3. Have the authors made all data underlying the findings in their manuscript fully available?

Reviewer #1: No

Reviewer #2: Yes

Reviewer #3: Yes

Reviewer #4: No

Reviewer #5: Yes

Reviewer #6: No

4. Is the manuscript presented in an intelligible fashion and written in standard English?

Reviewer #1: Yes

Reviewer #2: Yes

Reviewer #3: Yes

Reviewer #4: Yes

Reviewer #5: Yes

Reviewer #6: Yes

5. Review Comments to the Author

Reviewer #1: Jae-Chan Ryu and coworkers evaluated the relationship of inpatient blood pressure variability and CKD risk category with early neurological deterioration in patients with mild stroke. They report intriguing findings, but i have several reservations and suggestions.

Major comments:

————————

The Title and several additional phrases are misleading in that not “Renal Function” is studied but rather CKD risk categories. The authors must choose whether studying relationships with kidney function (e.g. eGFR, serum creatinine) or CKD risk categories.

In my opinion, BP variability should not by analyzed outside the context of blood pressure. the SD of BP, and to a lesser extent the CoV of BP, are both dependent on the blood pressure mean. Thus, SBP (or DBP) should be included in all models, alongside the BP variability index (SD or CoV, and the authors should consider also average real variability [ARV] and variability independent of the mean [VIM]).

When hypothesizing that the association between END and BPV is conditional upon CKD risk category, an appropriate interaction term must be included and tested in the statistical models (e.g. END ~ SD(SBP)*[CKD risk category]). It is not sufficient to claim that the association was only significant in a certain group and not the other.

Information about “medications stopped” should be presented and included in the models, as this clinical management decision (please cite an appropriate justification) surely increases BP variability.

Minor comments:

————————

Line 29 “estimated glomerulus filtration ratio (eGFR)” - glomerulus should be glomerular and ratio should be rate.

Lines 133-134 please also present ranges and p-value.

Line 143 - neurological severity on admission was LOWER in the highest CKD risk group according to the respective table.

Figure 1 is intriguing, but some values may be concealed due to the semi-3D nature. please provide all values in a supplementary table.

Table 2 - albuminuria should be presented as median and 25th-75th percentile.

Reviewer #2: Interesting study evaluating the effects of BPV on END in patients with and without CKD.In patients with minor stroke, study does show age as well as BP along with CKD affect neurological outcomes. Advanced CKD patients usually have atherosclerotic diseases that might be confounding factor.Nevertheless an interesting article to prove CKD is an important risk factor for stroke outcomes.Would consider authors to comment on whether the effect of ckd on END can be independent of BPV.

Reviewer #3: The authors revealed that the association between END and BPM parameters differs according to renal function in minor ischemic stroke; BPV was associated with END in patients with renal impairment, but less in those with normal renal function. These findings are interesting, and the manuscript is well written, hence the article would have potential for acceptation. Although several points should be described clearly.

I basically agree with your data and conclusion, but there were some discrepancies in patients with high risk levels of CKD patients. These data were seen to be severe outcomes in patients with moderate risk than high risk. Could the authors explain this inverted phenomenon?

When the authors confirmed the creatinine levels and determined the CKD stage? The creatinine levels may increase at the timing of admission (due to dehydration or some reason). Could the authors describe the timing for measurement the creatinine levels? Moreover, if the authors determined the CKD stage at admission, could the authors show the change of creatinine levels?

Could the authors show the medication for antihypertensive agents and antithrombotic agents at the time of admission? The class effect of antihypertensive drug on prevention for stroke was observed in the recent report (Zhu et al. Cochrane Database of Systematic Reviews 2022, Issue 1. Art. No.: CD003654.).

Were there differences for treatment for stroke between patients with CKD and non-CKD?

Because I could not realize the data of CKD G3bA3 in Figure 1, could the author show the data of Figure 1 not only 3D graph but also table?

Reviewer #4: The manuscript entitled of “Blood pressure variability (BPV) and early neurological deterioration (END) according to renal function in minor ischemic stroke patients” used a single center, retrospective data to analysis the effect of renal impairment on blood pressure and on early neurological deterioration in minor ischemic stroke cases. The authors found that there is a significant association of BPV to EDV in renal impairment group but not in the normal renal function group. However, several issue need to be addressed before further consideration.

Comments:

1. The author provided “CKD” related references in the Introduction part and cited the KDIGO guideline to classified the study subjects into 4 “risk groups” (low risk, moderate risk, high risk, and very high risk) in the method part. The low risk group is depicted as “normal renal function” in the method part. In the result part, the author used the term of “impair renal function” to include the moderate, high, and very high risk groups. And all these term were used in the discussion portion.

My concern is that these complicating terminology may confuse the readers. For example, a subject with eGFR more than 60 ml/min/m2 without albuminuria is not a CKD case by the KDIGO definition. The subjects of ”low risk” or “normal renal function” in this study maybe are non-CKD cases.

For CKD cases, the disease severity are usually classified into CKD stage I to V according to eGFR. However in this study, the author used the risk predicting grouping. To be noticed, the KDIGO guideline do provide evidence-based risk grouping but the classification varies regarding different outcomes. Therefore, I will suggest the method part should include clear definition of the above terminology, criteria for risk grouping, and the data for grouping should consider to be provided as a supplementary table.

2. Is the “stroke mechanism” at line 144, 173, of the Result part equals to the parameter of “subtypes of stroke”? I will suggest more deliberation or careful statement regarding the issue.

3. It is confusing that the author used both “SBP SD, SBPV SD", “SBP Cov, SBPV CoV” to name a few in tables, and in different sentences of results and discussion. Do they mean the same data? Or they were generated by different calculating algorithm? Please ensure the methodology and use consistent term. In addition, the BP variation is the key issue of this article. We could not tell if the BPV come from all BP records from every subjects of a group, or each individual’s BPV was in consideration in the analysis. Please include the rationale and citing the calculation method in the Method part.

Reviewer #5: In this single-center retrospective cohort study, the authors investigated whether the relationship between END after minor stoke and parameters of BPV was different between with and without impaired kidney function. Several BPV parameters were significantly associated with END in the subgroup with moderate to very high-risk CKD prognosis, but not in the subgroup with low risk (non-CKD) according to KDIGO 2012 classification.

1. The authors described that they evaluated parameters of BPV using the recorded blood pressure (BP) data during the first 5 days after admission. However, END was determined according to the NIHSS score change during 3 days after admission and BPV parameters in the END group were evaluated ignoring BP data after the diagnosis of END. Therefore, the duration obtaining BP data to evaluate BPV would be different between END and non-END groups. I think the period during which BP data were obtained to evaluate BPV parameters should be standardized to the first 3 days in both END and non-END patients.

2. In Table 3, results of multivariate logistic regression analyses in the normal and impaired renal function subgroups were shown. Please show the results of comparison of clinical parameters same as shown in Table 2 in the impaired renal function subgroup. In this subgroup, were there any clinical factors with significant difference except for age, NIHSS, and BPV parameters between patients with and without END? If there are factors which are significantly different between patients with and without END, they should be included in the multivariate logistic regression analysis.

3. Table S1 showed the results of multivariate logistic regression analysis and the authors argued that moderate and very high risk groups in the KDIGO classification were independently associated with END. Please discuss the possible reasons why only moderate and very high-risk group, but not high-risk group, were associated with END.

4. Figure 1 showed the distribution of percentage of END in each category of KDIGO classification. Very-high risk group looked like to have high prevalence of END, however, because the number of patients seemed to be quite different between categories and there were only 17 patients in the very-high risk group, Figure 1 might be misleading for the readers. Not only percentage but also absolute number of patients in each KDIGO risk category should be shown.

5. The described definition of END described in the Methods section seemed to be slightly different from that in the abstract. Which definition was true in your study?

6. Table 1 showed patient characteristics in this study. Type of medication for hypertension and use of anti-platelet agents should be also shown.

7. In the Results and Discussion section, the terms of “SBPV” and “DBPV” were used but they would be SBP and DBP.

Reviewer #6: The article Blood Pressure Variability and Early Neurological Deterioriation according to Renal Function in Minor Ischemic Stroke Patients by Ryu J.-C. tries to confirm the interrelationship between kidney function and brain damage. The authors retrospectively analyzed data of patients who were admitted hospital with acute minor ischemic stroke.

Although the study's hypothesis sounds promising, the approach to the available data has some significant limitations.

1. Introduction – should be improved. Indeed, kidney failure and brain damage share similar pathophysiological mechanisms that affect microcirculation (you mention some sentences in the discussion), and it is a result of vascular aging and other vascular risk factors that you mention (diabetes, hypertension). Moreover, emerging evidence shows that blood pressure variability is associated with arterial/aortic stiffness (a proxy of vascular age) that alters brain microcirculation and increases permeability in the blood-brain barrier, leading to brain damage.

2. Methods and Results:

a. The flow chart of the participant inclusion/exclusion would increase readability

b. Even though you acknowledge as a limitation that you had only one measurement of UACR, it is a considerable drawback of the study. Based on this single measurement, you then divide patients into risk groups. Additionally, you don’t mention the anthropometrical measurements of the patients. The thing is that obese individuals seem to have higher creatinine excretion in the urine and, therefore, false low UACR.

c. Blood pressure measurements – was it one measurement every 6 to 8 hours? Usually, standardized blood pressure should be performed, meaning three values in a row should be taken, and the average BP should be calculated.

d. You have divided patients into groups according to the KDIGO classification and had only 22 patients and 17 patients in high-risk and very high-risk groups, respectively. These numbers are too low to make conclusions about trends among groups. What is noteworthy is that you had 30 (30.9%) patients in the moderate CKD group with END, while only 8 (or 47.7%) patients with END were in the very high-risk group. I suppose you could try to analyze eGFR as a continuous variable without classifying it and show the connection between END and BPV.

e. Prognostic factors of END – the main limitation of this analysis is again too small sample numbers in high and very high-risk groups to conclude.

f. Table 3, in logistic regression, the BPV should be adjusted for diabetes even though diabetes was not significant in univariate analysis. Diabetes causes autonomic dysfunction that per se affects blood pressure variability. Hypertension and sex should also be included as covariates in logistic regression.

3. Discussion – you claim that “among those patients classified as very high risk, near half showed END.” It sounds like a solid and vital statement. However, you refer only to 8 patients from 17 in the very high-risk group. I suppose that you should tone down this statement.

---

## [Author Response · Author response to Decision Letter 0]

19 Aug 2022

All responses are included in the Response to Reviewers.

---

## [Editor Report · Decision Letter 1]

24 Aug 2022

Blood Pressure Variability and Early Neurological Deterioration according to the Chronic Kidney Disease Risk Categories in Minor Ischemic Stroke Patients

PONE-D-22-18870R1

Dear Dr. Kim,

We’re pleased to inform you that your manuscript has been judged scientifically suitable for publication and will be formally accepted for publication once it meets all outstanding technical requirements.

Kind regards,

Donovan Anthony McGrowder, PhD., MA., MSc

Academic Editor

PLOS ONE

Additional Editor Comments:

Dear Dr. Kim,

The manuscript entitled “Blood Pressure Variability and Early Neurological Deterioration according to the Chronic Kidney Disease Risk Categories in Minor Ischemic Stroke Patients” was revised in accordance with the reviewers’ comments and is provisionally accepted pending final checks for formatting and technical requirements.

Regards,

Dr. Donovan McGrowder (Academic Editor)

---

## [Editor Report · Acceptance letter]

29 Aug 2022

PONE-D-22-18870R1 

Blood Pressure Variability and Early Neurological Deterioration according to the Chronic Kidney Disease Risk Categories in Minor Ischemic Stroke Patients 

Dear Dr. Kim:

I'm pleased to inform you that your manuscript has been deemed suitable for publication in PLOS ONE. Congratulations! Your manuscript is now with our production department. 

Kind regards, 

on behalf of

Dr. Donovan Anthony McGrowder 

Academic Editor

PLOS ONE